# Abductive Reasoning in Logical Credal Networks

**Radu Marinescu**
IBM Research, Ireland
radu.marinescu@ie.ibm.com

**Junkyu Lee**
IBM Research, USA
junkyu.lee@ibm.com

**Debarun Bhattacharjya**
IBM Research, USA
debarunb@us.ibm.com

**Fabio Cozman**
Universidade de São Paulo, Brazil
fgcozman@usp.br

**Alexander Gray**
Centaur AI Institute, USA
alexander.gray@centaurinstitute.org

## Abstract

Logical Credal Networks or LCNs were recently introduced as a powerful probabilistic logic framework for representing and reasoning with imprecise knowledge. Unlike many existing formalisms, LCNs have the ability to represent cycles and allow specifying marginal and conditional probability bounds on logic formulae which may be important in many realistic scenarios. Previous work on LCNs has focused exclusively on marginal inference, namely computing posterior lower and upper probability bounds on a query formula. In this paper, we explore abductive reasoning tasks such as solving MAP and Marginal MAP queries in LCNs given some evidence. We first formally define the MAP and Marginal MAP tasks for LCNs and subsequently show how to solve these tasks exactly using search-based approaches. We then propose several approximate schemes that allow us to scale MAP and Marginal MAP inference to larger problem instances. An extensive empirical evaluation demonstrates the effectiveness of our algorithms on both random LCN instances as well as LCNs derived from more realistic use-cases.

## 1 Introduction

Probabilistic logic which combines probability and logic in a principled manner has emerged over the past decades as a unified representational and reasoning framework capable of dealing effectively with complex real-world applications that require efficient handling of uncertainty and compact representations of domain expert knowledge [1, 2, 3, 4, 5, 6, 7, 8, 9, 10]. Logical Credal Networks or LCNs [11] were introduced recently as a probabilistic logic designed for representing and reasoning with imprecise knowledge. Unlike many existing probabilistic logics, LCNs have the ability to represent cycles (e.g., feedback loops) as well as allow specifying marginal and conditional probability bounds on logic formulae which may be important in many realistic usecases.

Up until now, the work on LCNs has focused exclusively on marginal inference, i.e. efficiently computing posterior lower and upper probability bounds on a query formula. However, *abductive reasoning* tasks such as explaining the evidence observed in an LCN are equally important in many real-world applications. In probabilistic graphical models, these tasks are commonly known as MAP and Marginal MAP (MMAP) inference and have received extensive attention over the past decades [12, 13]. They are typically tackled efficiently with dynamic programming (e.g., variable elimination) or heuristic search (e.g., depth-first branch and bound) based algorithms [13, 14, 15, 16].

38th Conference on Neural Information Processing Systems (NeurIPS 2024).

**Contribution.** In this paper, we consider solving MAP and Marginal MAP inference queries in LCNs. Unlike in graphical models, an LCN encodes a set of probability distributions over its interpretations. Therefore, a complete or a partial explanation of the evidence which represents a complete or a partial truth assignment to the LCN's propositions may correspond to more than one distribution. Our work builds on very recent work on Marginal MAP inference for credal networks, a class of probabilistic graphical models that allow reasoning with imprecise probabilities [17]. We formally introduce the MAP and Marginal MAP tasks for LCNs as finding a complete or a partial truth assignment to the LCN's propositions with maximum *lower* (respectively, *upper*) probability, given some evidence in the LCN. We show how to evaluate such MAP assignments using exact marginal inference for LCNs and, subsequently, propose several search schemes based on depth-first search, limited discrepancy search and simulated annealing to solve these tasks in practice. We then extend a recent message-passing scheme for approximate marginal inference in LCNs [18] to handle effectively the MAP and MMAP inference tasks in LCNs as well as adapt the limited discrepancy search and simulated annealing methods to use an approximate evaluation of the MAP assignments during search. We experiment and evaluate our proposed exact and approximate algorithms on several classes of LCNs including random as well as more realistic LCN instances. Our results show that the search methods based on exact evaluation of the MAP assignments are limited to solving small size problems in practice, while the approximate message-passing scheme and, to some extent, the approximate search-based methods can scale to much larger problem instances. This is important because it allows us to tackle practical problems involving hundreds and possibly many thousands of propositions. The supplementary material includes additional details and experiments.

## 2 Background

We provide next a brief overview of basic concepts about LCNs and marginal inference in these models. Throughout the paper we will use the following notations. Logical propositions are denoted by uppercase letters (e.g., $A, B, C, ...$) while for sets of propositions we use boldfaced uppercase letters (e.g., $\mathbf{A}, \mathbf{B}, \mathbf{C}, ...$). Truth assignments to propositions (i.e., *literals*) are denoted by either lowercase or uppercase letters, namely we use $a$ or $A$ to indicate that proposition $A$ holds true, and $\neg a$ or $\neg A$ if $A$ is false. Sets of literals are denoted by boldfaced lowercase letters (e.g., $\mathbf{a}, \mathbf{b}, \mathbf{c}, ...$).

### 2.1 Logical Credal Networks

A Logical Credal Network (LCN) [11] is defined by a tuple $\mathcal{L} = \langle \mathbf{A}, \mathcal{C} \rangle$, where $\mathbf{A} = \{A_1, \ldots, A_n\}$ is a set of propositions (or atoms), and $\mathcal{C}$ is a set of probability labeled sentences (or constraints) having the following two forms:

$$\alpha \leq P(\phi) \leq \beta \tag{1}$$

$$\alpha \leq P(\phi|\varphi) \leq \beta \tag{2}$$

Here, $\phi$ and $\varphi$ are arbitrary propositional logic formulae[1] involving propositions in $\mathcal{A}$ and logical connectives such as negation, disjunction and conjunction, and $0 \leq \alpha \leq \beta \leq 1$ are lower and upper probability bounds, respectively.

An LCN is associated with *primal graph* which is a directed graph $G$ containing *formula nodes* and *proposition nodes*, as well as directed edges from each proposition node in a formula $\phi$ to the formula node representing $\phi$ (for type 1 sentences), and directed edges from each of the proposition nodes in $\varphi$ to $\varphi$, a directed edge from $\varphi$ to $\phi$, and bi-directed edges from $\phi$ to the proposition nodes in $\phi$, respectively (for type 2 sentences) [11]. A *parent* of a proposition $A$ in $G$ is a proposition $B$ such that there is a directed path in $G$ from $B$ to $A$ in which all intermediate nodes are formulae. A *descendant* of a proposition $A$ in $G$ is a proposition $B$ such that there is a directed path in $G$ from $A$ to $B$ in which no intermediate node is a parent of $A$ [11].

An LCN is endowed with a *Local Markov Condition* (LMC) where a proposition node $A$ is independent, given its parents, of all proposition nodes that are not $A$ itself nor descendants of $A$ nor parents of $A$ [11]. Therefore, an LCN represents a set of probability distributions over all interpretations of its formulae that satisfy the constraints represented by the type (1) and (2) sentences as well as the constraints induced by the independence relations given by the local Markov condition [11].

---

[1]The original definition of LCNs allows for relational structures and first-order logic formulae, but their semantics is obtained by grounding on finite domains thus yieling a propositional LCN [11].

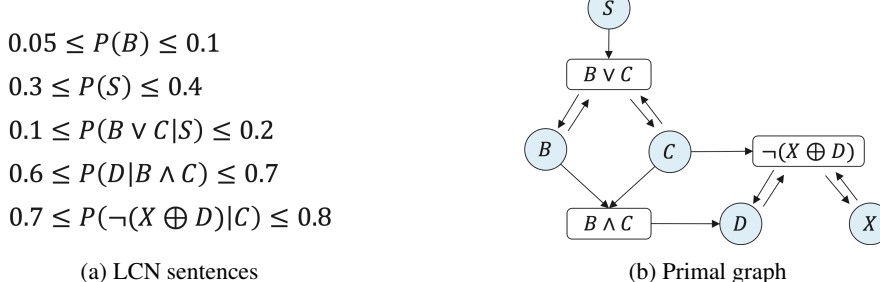

$$0.05 \leq P(B) \leq 0.1$$
$$0.3 \leq P(S) \leq 0.4$$
$$0.1 \leq P(B \vee C|S) \leq 0.2$$
$$0.6 \leq P(D|B \wedge C) \leq 0.7$$
$$0.7 \leq P(\neg(X \oplus D)|C) \leq 0.8$$

(a) LCN sentences          (b) Primal graph

Figure 1: A simple LCN and its primal graph.

**Example 1.** *Figure 1 describes a simple LCN whose sentences shown in Figure 1a state that: Bronchitis (B) is more likely than Smoking (S); Smoking may cause Cancer (C) or Bronchitis; Dyspnea (D) or shortness of breadth is a common symptom for Cancer and Bronchitis; in case of Cancer we have either a positive X-Ray result (X) and Dyspnea, or a negative X-Ray and no Dyspnea. Figure 1b shows the primal graph where the formula and proposition nodes are displayed as rectangles and shaded circles, respectively.*

## 2.2 Marginal Inference in Logical Credal Networks

Given an LCN $\mathcal{L}$ with $n$ propositions, the *marginal inference* task is to compute lower and upper bounds on the posterior probability $P(\psi)$ of a query formula $\psi$, which we denote by $\underline{P}(\psi)$ and $\overline{P}(\psi)$, respectively. This is achieved by solving a non-linear program given by Equations (3)–(8) and defined by a set of non-negative real-valued variables representing the probabilities of $\mathcal{L}$'s interpretations, a set of linear constraints derived from $\mathcal{L}$'s sentences, a set of non-linear constraints corresponding to the independence

$$\sum_{i=1}^{m} p_i = 1 \tag{3}$$

$$p_i \geq 0, \forall i = 1, \ldots, m \tag{4}$$

$$\alpha \leq \vec{I}_\phi \odot \vec{p} \leq \beta \tag{5}$$

$$\alpha \cdot \vec{I}_\varphi \odot \vec{p} \leq \vec{I}_{\phi \wedge \varphi} \odot \vec{p} \leq \beta \cdot \vec{I}_\varphi \odot \vec{p} \tag{6}$$

$$(\vec{I}_a \odot \vec{p}) \cdot (\vec{I}_b \odot \vec{p}) - (\vec{I}_c \odot \vec{p}) \cdot (\vec{I}_d \cdot \vec{p}) = 0 \tag{7}$$

$$\text{minimize/maximize } \vec{I}_\psi \odot \vec{p} \tag{8}$$

assumptions given by the local Markov condition, and a linear objective function encoding the query $P(\psi)$ which is minimized and maximized to yield the desired bounds. More specifically, let $\vec{p} = (p_1, \ldots, p_m)$ be the vector of real-valued variables representing the probabilities of $\mathcal{L}$'s interpretations, where $m = 2^n$, and let $\vec{I}_\phi = (a_1^\phi, \ldots, a_m^\phi)$ be a binary vector, called an *indicator vector*, such that $a_i^\phi$ is 1 if formula $\phi$ is true in the $i$-th interpretation and 0 otherwise. Since the probability of a formula $\phi$ is the sum of the probabilities of the interpretations in which $\phi$ is true, we can write $P(\phi)$ as $\vec{I}_\phi \odot \vec{p}$ where $\odot$ is the dot-product of two vectors. Therefore, Equations (3) and (4) ensure that $\vec{p}$ is a valid probability distribution, Equations (5) and (6) encode the type (1) and (2) sentences in $\mathcal{L}$ while Equation 7 encodes the conditional independencies of the form $P(X_j|\mathbf{S}_j, \mathbf{T}_j) = P(X_j|\mathbf{S}_j)$, where $X_j$ is a proposition, $\mathbf{S}_j = \{S_{j1}, \ldots, S_{jk}\}$ and $\mathbf{T}_j = \{T_{j1}, \ldots, T_{jl}\}$ are $X_j$'s parents and non-descendants in the primal graph of $\mathcal{L}$, $\vec{I}_\phi$ and $\vec{I}_{\phi \wedge \varphi}$ are the indicator vectors for formulae $\phi$ and $\phi \wedge \varphi$ involved in $\mathcal{L}$'s sentences, and $\vec{I}_a$, $\vec{I}_b$, $\vec{I}_c$ and $\vec{I}_d$ are the indicator vectors corresponding to the formulae $a = (x_j \wedge s_{j1} \wedge \cdots \wedge s_{jk} \wedge t_{j1} \wedge \cdots \wedge t_{jl})$, $b = (s_{j1} \wedge \cdots \wedge s_{jk})$, $c = (x_j \wedge s_{j1} \wedge \cdots \wedge s_{jk})$, and $d = (s_{j1} \wedge \cdots \wedge s_{jk} \wedge t_{j1} \wedge \cdots \wedge t_{jl})$, respectively (see also [11] for more details).

## 3 MAP and Marginal MAP Inference in LCNs

Maximum A Posteriori (MAP) and Marginal MAP (MMAP) inference are well known abductive reasoning tasks in probabilistic graphical models such as Bayesian networks and Markov networks [12, 13, 14, 15, 16]. Specifically, the MAP task calls for finding a complete assignment to all variables having maximum probability, given the evidence. Marginal MAP generalizes MAP and looks for a partial variable assignment that has maximum marginal probability, given the evidence. MAP

and MMAP inference tasks appear in many real-world applications such as diagnosis, abduction and explanation and are typically tackled with dynamic programming (e.g., variable elimination) or heuristic search (e.g., depth-first branch and bound) based algorithms [13, 14, 15, 16].

In this section, we present our novel approach for solving the MAP and Marginal MAP inference tasks in Logical Credal Networks. Unlike in graphical models, a (partial) variable assignment (or interpretation) in an LCN may correspond to more than one distribution. Therefore, we begin by formally defining two MAP and MMAP inference tasks for LCNs, called *maximin MAP* (resp. *maximin MMAP*) and *maximax MAP* (resp. *maximax MMAP*). Subsequently, we develop several exact and approximation schemes for solving these tasks efficiently in practice.

## 3.1 The MAP and Marginal MAP Tasks in LCNs

Let $\mathcal{L} = \langle \mathbf{A}, \mathcal{C} \rangle$ be an LCN with $n$ propositions and let $\mathbf{E} = \{E_1, \ldots, E_k\} \subseteq \mathbf{A}$ be a subset of $k$ propositions, called *evidence*, for which the truth values $\mathbf{e} = \{e_1, \ldots, e_k\}$ are known. Let $\mathbf{Y} = \{Y_1, \ldots, Y_m\} \subseteq \mathbf{A} \setminus \mathbf{E}$ be a subset of $m$ propositions called *MAP propositions*. A truth assignment to $\mathbf{Y}$ is is called a *MAP assignment* and is denoted by $\mathbf{y} = \{y_1, \ldots, y_m\}$, respectively. Clearly, if $\mathbf{Y} = \mathbf{A} \setminus \mathbf{E}$ (i.e., $m = n - k$) then we have a MAP task, otherwise we have a MMAP task (i.e., $m < n - k$). The *maximin* and *maximax* MAP/MMAP tasks are defined as follows:

**Definition 1** (maximin). *Given an LCN $\mathcal{L}$ with $n$ propositions, evidence $\mathbf{e}$, and MAP propositions $\mathbf{Y}$, the* maximin *MAP (or* maximin *MMAP if $m < n - k$) task is finding a truth assignment $\mathbf{y}^*$ to $\mathbf{Y}$ having maximum lower probability, given evidence $\mathbf{e}$, namely:*

$$\mathbf{y}^* = \underset{\mathbf{y} \in \Omega(\mathbf{Y})}{\operatorname{argmax}} \underline{P}(\psi_{\mathbf{y} \wedge \mathbf{e}}) \tag{9}$$

*where $\Omega(\mathbf{Y})$ is the set of all truth assignments to the MAP propositions, and $\psi_{\mathbf{y} \wedge \mathbf{e}} = y_1 \wedge \cdots \wedge y_m \wedge e_1 \wedge \cdots \wedge e_k$ is the conjunction of the literals in $\mathbf{y}$ and $\mathbf{e}$, respectively.*

**Definition 2** (maximax). *Given an LCN $\mathcal{L}$ with $n$ propositions, evidence $\mathbf{e}$, and MAP propositions $\mathbf{Y}$, the* maximax *MAP (or* maximax *MMAP if $m < n - k$) task is finding a truth assignment $\mathbf{y}^*$ to $\mathbf{Y}$ having maximum upper probability, given evidence $\mathbf{e}$, namely:*

$$\mathbf{y}^* = \underset{\mathbf{y} \in \Omega(\mathbf{Y})}{\operatorname{argmax}} \overline{P}(\psi_{\mathbf{y} \wedge \mathbf{e}}) \tag{10}$$

*where $\Omega(\mathbf{Y})$ is the set of all truth assignments to the MAP propositions, and $\psi_{\mathbf{y} \wedge \mathbf{e}} = y_1 \wedge \cdots \wedge y_m \wedge e_1 \wedge \cdots \wedge e_k$ is the conjunction of the literals in $\mathbf{y}$ and $\mathbf{e}$, respectively.*

## 3.2 Search Algorithms Using Exact MAP Assignment Evaluations

We present next three search-based schemes for solving the MAP and MMAP tasks in LCNs. These methods employ different search strategies for exploring the search space defined by the MAP propositions while evaluating exactly each complete or partial MAP assignment.

**Exact Evaluation of a MAP Assignment.**   Clearly, computing the lower and upper probabilities $\underline{P}(\psi_{\mathbf{y} \wedge \mathbf{e}})$ and $\overline{P}(\psi_{\mathbf{y} \wedge \mathbf{e}})$ of a MAP assignment $\mathbf{y}$ given evidence $\mathbf{e}$ can be done easily by minimizing and, respectively maximizing the non-linear program defined by Equations (3)–(8), where the query formula is the conjunction of positive or negative literals in $\mathbf{y}$ and $\mathbf{e}$, namely $\psi_{\mathbf{y} \wedge \mathbf{e}} = y_1 \wedge \cdots \wedge y_m \wedge e_1 \wedge \cdots \wedge e_k$. Therefore, evaluating a MAP assignment in case of both MAP and Marginal MAP inference in LCNs is quite difficult as it involves solving a maginal inference problem for LCNs which is know to be NP-hard [11]. This is in contrast with graphical models where, at least for MAP inference, the evaluation of a MAP assignment is linear in the number of variables [13].

**Example 2.** *For illustration, consider the LCN example from Figure 1 and assume that we have evidence $\mathbf{e} = \{x, \neg s\}$, namely a patient has a positive X-Ray result ($X = x$) and is not smoking ($S = \neg s$). The MAP propositions in this case are $\mathbf{Y} = \{B, C, D\}$ and the MAP assignment $\mathbf{y} = (b, \neg c, \neg d)$ corresponds to the query formula $\psi_{\mathbf{y} \wedge \mathbf{e}} = b \wedge \neg c \wedge \neg d \wedge x \wedge \neg s$. The lower and upper probabilities $\underline{P}(\psi_{\mathbf{y} \wedge \mathbf{e}})$ and $\overline{P}(\psi_{\mathbf{y} \wedge \mathbf{e}})$ of the MAP assignment are 9.9e-09 and 0.1, respectively.*

**Algorithm 1** Depth-First Search for MAP and Marginal MAP Inference in LCNs

1: **procedure** DFS($\mathcal{L} = \langle \mathbf{A}, \mathcal{C} \rangle$, $\mathbf{E} = \mathbf{e}$, $\mathbf{Y}$)
2:   initialize $\mathbf{y}^* \leftarrow \emptyset$, $best \leftarrow -\infty$
3:   SEARCH($\emptyset$, $\mathbf{Y}$)
4:   **return** $\mathbf{y}^*$
5: **procedure** SEARCH($\mathbf{y}$, $\mathbf{Y}$)
6:   **if** $size(\mathbf{y}) == size(\mathbf{Y})$ **then**
7:     **if** maximin **then**
8:       $score(\mathbf{y}) \leftarrow \underline{P}(\psi_{\mathbf{y} \wedge \mathbf{e}})$
9:     **else**
10:       $score(\mathbf{y}) \leftarrow \overline{P}(\psi_{\mathbf{y} \wedge \mathbf{e}})$
11:     **if** $score(\mathbf{y}) > best$ **then**
12:       $\mathbf{y}^* \leftarrow \mathbf{y}$, $best \leftarrow score(\mathbf{y})$
13:   **else**
14:     select unassigned proposition $Y_i \in \mathbf{Y}$
15:     **for all** values $y \in \{y_i, \neg y_i\}$ **do**
16:       $\mathbf{y} \leftarrow \mathbf{y} \cup \{Y_i = y\}$
17:       SEARCH($\mathbf{y}$, $\mathbf{Y}$)

---

**Algorithm 2** Limited Discrepancy Search for MAP and Marginal MAP Inference in LCNs

1: **procedure** LDS($\mathcal{L} = \langle \mathbf{A}, \mathcal{C} \rangle$, $\mathbf{E} = \mathbf{e}$, $\mathbf{Y}$, $\delta$)
2:   initialize $\mathbf{y}_0$ randomly and let $\mathbf{y}^* \leftarrow \mathbf{y}_0$
3:   $best \leftarrow score(\mathbf{y}^*)$
4:   **for all** $\theta = 1 \ldots \delta$ **do**
5:     SEARCH($\mathbf{y}^*$, $\mathbf{Y}$, $\theta$, $1$)
6:   **return** $\mathbf{y}^*$, $best$
7: **procedure** SEARCH($\mathbf{y}$, $\mathbf{Y}$, $\theta$, $i$)
8:   **if** $\theta == 0$ or $i > |\mathbf{Y}|$ **then**
9:     **if** maximin **then**
10:       $score(\mathbf{y}) \leftarrow \underline{P}(\psi_{\mathbf{y} \wedge \mathbf{e}})$
11:     **else**
12:       $score(\mathbf{y}) \leftarrow \overline{P}(\psi_{\mathbf{y} \wedge \mathbf{e}})$
13:     **if** $score(\mathbf{y}) > best$ **then**
14:       $\mathbf{y}^* \leftarrow \mathbf{y}$, $best \leftarrow score(\mathbf{y})$
15:   **else**
16:     **for all** values $y \in \{y_i, \neg y_i\}$ **do**
17:       **if** $\mathbf{y}[i] == y$ **then**
18:         $\mathbf{z} \leftarrow$ SEARCH($\mathbf{y}$, $\mathbf{Y}$, $i+1$, $\theta$)
19:       **else**
20:         $\mathbf{y}' \leftarrow \mathbf{y}$; $\mathbf{y}'[i] \leftarrow y$
21:         $\mathbf{z} \leftarrow$ SEARCH($\mathbf{y}'$, $\mathbf{Y}$, $i+1$, $\theta-1$)
22:     **return** $\mathbf{z}$

---

**Depth-First Search.** Our first approach for solving the MAP and MMAP tasks, called DFS, is described by Algorithm 1. It takes as input an LCN $\mathcal{L} = \langle \mathbf{A}, \mathcal{C} \rangle$, evidence $\mathbf{E} = \mathbf{e}$ and a set of MAP propositions $\mathbf{Y} \subseteq \mathbf{A} \setminus \mathbf{E}$ and outputs the optimal MAP assignment $\mathbf{y}^*$. The method conducts a *depth-first search* over the space of partial assignments to the MAP propositions, and, for each complete MAP assignment $\mathbf{y}$ computes its score as the exact lower probability $\underline{P}(\psi_{\mathbf{y} \wedge \mathbf{e}})$ for maximin tasks, and respectively, the upper probability $\overline{P}(\psi_{\mathbf{y} \wedge \mathbf{e}})$ for maximax tasks, given the evidence $\mathbf{e}$. This way, the optimal solution $\mathbf{y}^*$ corresponds to the MAP assignment with the highest score.

**Theorem 1** (complexity). *Given an LCN $\mathcal{L} = \langle \mathbf{A}, \mathcal{C} \rangle$ with $n$ propositions, evidence $\mathbf{E} = \mathbf{e}$ and MAP propositions $\mathbf{Y} \subseteq \mathbf{A} \setminus \mathbf{E}$, algorithm DFS is sound and complete. The time and space complexity of the algorithm is $O(2^{m+2^n})$ and $O(2^n)$, respectively, where $m$ is the number of MAP propositions.*

**Example 3.** *Consider again the LCN from Figure 1 with evidence $\mathbf{e} = \{x, \neg s\}$. In this case, the exact maximin MAP assignment found by algorithm DFS is $\mathbf{y}^* = \{\neg b, c, d\}$ with value 9.99e-09, while the exact maximax MAP assignment is $\mathbf{y}^* = \{\neg b, \neg c, d\}$ with value 0.7, respectively.*

**Limited Discrepancy Search.** Our second approach for MAP and MMAP inference in LCNs uses Limited Discrepancy Search (LDS) [19, 20] to explore the search space and is described by Algorithm 2. Specifically, LDS is a depth-first search strategy that searches for new solutions by iteratively increasing the number of *discrepancy* values, where a discrepancy value indicates the maximum number of allowed variable-value assignment changes to an initial solution [19]. Function SEARCH (lines 7–22) performs the actual exploration of the search space limited by discrepancy $\theta$. If the selected truth value $y \in \{y_i, \neg y_i\}$ is different from the one corresponding to proposition $Y_i \in \mathbf{Y}$ at position $i$ in the assignment $\mathbf{y}$, $\theta$ is decremented to reduce the number of changes allowed to the remaining MAP propositions. Otherwise, the truth value for proposition $Y_i$ remains unchanged and the $\theta$ value is preserved. As before, complete MAP assignments are evaluated exactly (lines 9–12) and the best solution found so far is maintained (lines 13-14).

---

**Algorithm 3** Simulated Annealing for MAP and Marginal MAP Inference in LCNs

---

1: **procedure** SA($\mathcal{L} = \langle \mathbf{A}, \mathcal{C} \rangle$, $\mathbf{E} = \mathbf{e}$, $\mathbf{Y}$)
2:   initialize $\mathbf{y}_0$ randomly and let $\mathbf{y}^* \leftarrow \mathbf{y}_0$
3:   $best \leftarrow score(\mathbf{y}^*)$
4:   **for all** iterations $i = 1 \ldots N$ **do**
5:     set $\mathbf{y} \leftarrow \mathbf{y}^*$, $T \leftarrow T_{init}$
6:     **for all** flips $j = 1 \ldots M$ **do**
7:       let $\mathcal{N}$ be $\mathbf{y}$'s neighbors
8:       select random neighbor $\mathbf{y}' \in \mathcal{N}$
9:       $\Delta \leftarrow \log score(\mathbf{y}') - \log score(\mathbf{y})$
10:       **if** $\Delta > 0$ **then** $\mathbf{y} \leftarrow \mathbf{y}'$
11:       **else**
12:         sample randomly $p \in (0, 1)$
13:         **if** $p < e^{\frac{\Delta}{T}}$ **then** $\mathbf{y} \leftarrow \mathbf{y}'$
14:       **if** $score(\mathbf{y}) > best$ **then**
15:         $\mathbf{y}^* \leftarrow \mathbf{y}$, $best \leftarrow score(\mathbf{y})$
16:       $T \leftarrow T * \sigma$
17:   **return** $\mathbf{y}^*$

---

---

**Algorithm 4** Approximate MAP and Marginal MAP Inference in LCNs

---

1: **procedure** AMAP($\mathcal{L} = \langle \mathbf{A}, \mathcal{C} \rangle$, $\mathbf{E} = \mathbf{e}$, $\mathbf{Y}$)
2:   Create factor graph $\mathcal{F}$ of $\mathcal{L}$
3:   Apply the ARIEL scheme from [18] on $\mathcal{F}$
4:   **for all** MAP propositions $Y \in \mathbf{Y}$ **do**
5:     **if** maximin **then**
6:       $\underline{P}(y) = \max_{f \in N(Y)} l_{f \rightarrow Y}$
7:       $\underline{P}(\neg y) = 1 - \underline{P}(y)$
8:       **if** $\underline{P}(y) > \underline{P}(\neg y)$ **then** $\mathbf{y}^* \leftarrow \mathbf{y}^* \cup \{y\}$
9:       **else** $\mathbf{y}^* \leftarrow \mathbf{y}^* \cup \{\neg y\}$
10:     **else**
11:       $\overline{P}(y) = \min_{f \in N(Y)} u_{f \rightarrow Y}$
12:       $\overline{P}(\neg y) = 1 - \overline{P}(y)$
13:       **if** $\overline{P}(y) > \overline{P}(\neg y)$ **then** $\mathbf{y}^* \leftarrow \mathbf{y}^* \cup \{y\}$
14:       **else** $\mathbf{y}^* \leftarrow \mathbf{y}^* \cup \{\neg y\}$
15:   **return** $\mathbf{y}^*$

---

**Theorem 2** (complexity). *Given an LCN $\mathcal{L} = \langle \mathbf{A}, \mathcal{C} \rangle$ with $n$ propositions, evidence $\mathbf{E} = \mathbf{e}$ and MAP propositions $\mathbf{Y} \subseteq \mathbf{A} \setminus \mathbf{E}$, algorithm LDS is sound and complete. The time and space complexity of the algorithm is $O(2^{m+2^n})$ and $O(2^n)$, respectively, where $m$ is the number of MAP propositions.*

**Simulated Annealing.** The third approach for solving MAP and MMAP tasks in LCNs is described by Algorithm 3 and employs a form of stochastic local search known as Simulated Annealing (SA) [21] to explore the search space defined by the MAP propositions. The algorithm starts from an initial guess $\mathbf{y}$ as a truth assignment to the MAP propositions $\mathbf{Y}$, and iteratively tries to improve it by moving to a better neighbor $\mathbf{y}'$ that has a higher score. A *neighbor* $\mathbf{y}'$ of $\mathbf{y}$ is defined as a new assignment $\mathbf{y}'$ which results from changing the truth value of a single proposition $Y$ in $\mathbf{Y}$. At each step, the transition from the current state $\mathbf{y}$ to a neighboring state $\mathbf{y}'$ is decided probabilistically using an acceptance probability function $P(\mathbf{y}', \mathbf{y}, T)$ that depends on the scores of the two states as well as a global time-varying parameter $T$ called *temperature* which is decreased using a cooling schedule $\sigma < 1$ [21]. We chose $P(\mathbf{y}', \mathbf{y}, T) = e^{\frac{\Delta}{T}}$, where $\Delta = \log score(\mathbf{y}') - \log score(\mathbf{y})$.

**Theorem 3** (complexity). *Given an LCN $\mathcal{L} = \langle \mathbf{A}, \mathcal{C} \rangle$ with $n$ propositions, evidence $\mathbf{E} = \mathbf{e}$ and MAP propositions $\mathbf{Y} \subseteq \mathbf{A} \setminus \mathbf{E}$, the time and space complexity of algorithm SA is $O(N \cdot M \cdot 2^{2^n})$ and $O(2^n)$, respectively, where $N$ is the number of iterations and $M$ is the number of flips per iterations.*

### 3.3 Approximate MAP and Marginal MAP Inference

The main bottleneck in the proposed search algorithms is the exact evaluation of the MAP assignments which is computationally very expensive [11]. This limits the applicability of these methods to relatively small LCNs. Therefore, in order to be able to tackle larger LCNs, we extend a recent message-passing approximation scheme for marginal inference in LCNs [18] to solve the MAP and MMAP tasks in LCNs. Subsequently, we also adapt the limited discrepancy search and simulated annealing methods to use an approximate evaluation of the MAP assignments during search.

Algorithm 4 describes our message-passing based approximation scheme for MAP and MMAP inference in LCNs which we denote hereafter by AMAP. We build upon a recent scheme for approximate marginal inference in LCNs, called ARIEL [18], which propagates messages along the edges of a *factor graph* associated with the input LCN until convergence. The factor graph $\mathcal{F}$ of an

LCN $\mathcal{L}$ is a bi-partite graph the connects *proposition nodes* labeled by the propositions in $\mathcal{L}$ with *factor nodes* associated with sentences that involve the same set of propositions [18]. The messages propagated between the nodes of $\mathcal{F}$ are intervals representing lower and upper bounds on the marginal probabilities of $\mathcal{L}$'s propositions and are computed as follows: the message sent from a proposition to a factor node tightens these bounds based on the incoming messages from the factor nodes connected to it; the message sent from a factor to a proposition node computes new bounds by solving a local non-linear program defined by the factor's sentences and the constraints encoding the assumption that the factor's propositions are independent of each other and the marginal probabilities of the factor's propositions are within the bounds given by the incoming proposition-to-factor messages (see also [18] for more details). Upon convergence, the maximin MAP assignment $\mathbf{y}^*$ can be obtained as follows: for each MAP proposition $Y \in \mathbf{Y}$ we compute the tightest lower probability bound $\underline{P}(y)$ by maximizing the lower bound of all incoming factor-to-proposition messages to $Y$, and, subsequently, select $y$ as the most likely value assignment to $Y$ if $\underline{P}(y) > \underline{P}(\neg y)$ and $\neg y$ otherwise (for the maximax tasks we use the upper probability bounds $\overline{P}(y)$ and $\overline{P}(\neg y)$, respectively).

**Theorem 4** (complexity). *Given an LCN $\mathcal{L} = \langle \mathbf{A}, \mathcal{C} \rangle$ with $n$ propositions, evidence $\mathbf{E} = \mathbf{e}$ and MAP propositions $\mathbf{Y} \subseteq \mathbf{A} \setminus \mathbf{E}$, the time and space complexity of algorithm AMAP is $O(N \cdot M \cdot 2^{2^r})$ and $O(2^r)$, where $N$ is the number of iterations, $M$ bounds the number of factor-to-node messages per iteration and $r$ bounds the number of propositions in the factor nodes, respectively.*

### 3.4 Search Algorithms Based on Approximate MAP Evaluations

The main assumption behind algorithm AMAP is that all MAP propositions are independent of each other and therefore the solution $\mathbf{y}^*$ returned by AMAP is likely to correspond to a local maxima. One way to escape such a local optima and obtain a potentially better solution is to employ a search scheme based on either limited discrepancy search or simulated annealing that continues the exploration of the search space starting from $\mathbf{y}^*$. However, in order to scale to larger LCNs, we would like the search schemes to rely on an approximate rather than an exact evaluation of the MAP assignments.

**Approximate Evaluation of a MAP Assignment.** Estimating the lower and upper probabilities of a MAP assignment $\mathbf{y}$ can be done by approximate marginal inference on an *augmented* LCN as follows. Let $\mathcal{L} = \langle \mathbf{A}, \mathcal{C} \rangle$ be the input LCN and let $\mathbf{y} = (y_1, \ldots, y_m)$ be a MAP assignment to propositions $\mathbf{Y} = \{Y_1, \ldots, Y_m\}$ (for simplicity, we include the evidence $\mathbf{e}$ in $\mathbf{y}$). The *augmented* LCN $\mathcal{L}' = \langle \mathbf{A}', \mathcal{C}' \rangle$ is constructed by adding a set of auxiliary propositions $\mathbf{W} = \{W_1, \ldots, W_m\}$, one for each MAP proposition, and additional constraints of the following two forms: $P(W_1|Y_1)$ and $P(W_j|W_{j-1} \wedge Y_j)$, for all $2 \le j \le m$, such that $P(w_1|y_1) = 1$, $P(w_1|\neg y_1) = 0$, $P(w_j|w_{j-1} \wedge y_j) = 1$, $P(w_j|w_{j-1} \wedge \neg y_j) = 0$, $P(w_j|\neg w_{j-1} \wedge y_j) = 0$ and $P(w_j|\neg w_{j-1} \wedge \neg y_j) = 0$, respectively. Then, we can estimate $\underline{P}(\psi_{\mathbf{y}})$ and $\overline{P}(\psi_{\mathbf{y}})$, where $\psi_{\mathbf{y}} = y_1 \wedge \cdots \wedge y_m$, by computing the posterior marginals $\underline{P}(w_m)$ and $\overline{P}(w_m)$ in the augmented LCN $\mathcal{L}'$ using the method from [18].

**Limited Discrepancy Search and Simulated Annealing.** Our approximate LDS and SA based algorithms denoted by ALDS and ASA can be obtained from Algorithms 2 and 3 by replacing the $score(\mathbf{y})$ function with the approximate MAP evaluation scheme described above. These algorithms can start the search either from a random MAP assignment or from the solution found by algorithm AMAP. Finally, the time complexity of algorithms ALDS and ASA can be bounded by $O(2^{m+2^r})$ and $O(N \cdot M \cdot 2^{2^r})$, respectively, where $m$ is the number of MAP propositions, $N$ is the number of iterations used by ASA, $M$ is the maximum number of flips per iteration, and $r$ bounds the number of propositions in the factor nodes of the factor graph associated with the input LCN [18].

## 4 Experiments

In this section, we empirically evaluate the proposed exact and approximate schemes for MAP and MMAP inference in LCNs. All competing algorithms were implemented[2] in Python 3.10 and used the `ipopt` 3.14 solver [22] with default settings to handle the non-linear constraint programs. We ran all experiments on a 3.0GHz Intel Core processor with 128GB of RAM.

---

[2]The open-source implementation of LCNs is available at: https://github.com/IBM/LCN

Table 1: Results for MAP tasks obtained on small/large scale `polytree`, `dag`, and `random` LCNs. Average CPU time in seconds and number of problem instances solved. Time limit is 2 hours.

| size | exact MAP eval | | | | approx MAP eval | |
|---|---|---|---|---|---|---|
| $n$ | DFS | LDS(3) | SA | AMAP | ALDS(3) | ASA |
| polytree | | | | | | |
| 5 | 15.30 (10) | 26.07 (10) | 20.18 (10) | 2.87 (10) | 174.17 (10) | 188.27 (10) |
| 8 | 3246.28 (4) | 3072.18 (4) | 1199.51 (10) | 8.05 (10) | 1054.53 (10) | 518.18 (10) |
| 10 | - | - | - | 11.81 (10) | 2273.16 (10) | 813.30 (10) |
| 30 | - | - | - | 31.55 (10) | - | 3091.74 (10) |
| 50 | - | - | - | 52.30 (10) | - | 5324.71 (10) |
| 70 | - | - | - | 79.28 (10) | - | 7279.56 (10) |
| dag | | | | | | |
| 5 | 21.09 (10) | 15.66 (10) | 24.04 (10) | 5.54 (10) | 163.02 (10) | 156.34 (10) |
| 8 | 1633.38 (8) | 1958.16 (9) | 633.77 (10) | 13.05 (10) | 1339.71 (10) | 571.55 (10) |
| 10 | - | - | - | 15.55 (10) | 2903.05 (10) | 944.17 (10) |
| 30 | - | - | - | 49.94 (10) | - | 3593.71 (10) |
| 50 | - | - | - | 89.13 (10) | - | 5639.90 (10) |
| 70 | - | - | - | 132.34 (10) | - | 6093.28 (10) |
| random | | | | | | |
| 5 | 19.51 (10) | 17.56 (10) | 20.37 (10) | 5.26 (10) | 152.99 (10) | 143.60 (10) |
| 8 | 3152.57 (1) | 3209.54 (5) | 1226.88 (10) | 10.29 (10) | 954.46 (10) | 444.17 (10) |
| 10 | - | - | - | 12.21 (10) | 2150.27 (10) | 717.75 (10) |
| 30 | - | - | - | 40.54 (10) | - | 3335.14 (10) |
| 50 | - | - | - | 76.83 (10) | - | 5276.93 (10) |
| 70 | - | - | - | 105.70 (10) | - | 6059.57 (7) |

**Random LCNs.** We generated three classes of random LCNs with $n$ propositions $\{X_1, \ldots X_n\}$ and sentences of the following types: (a) $l \leq P(x_i) \leq u$, (b) $l \leq P(x_i|x_j) \leq u$, $i \neq j$ and (c) $l \leq P(x_i|X_j \wedge X_k) \leq u$, $i \neq j \neq k$, such that the corresponding primal graph is a `polytree`, a `dag` or a `random` graph. The type (c) sentences were generated for all truth values of propositions $X_j$ and $X_k$, namely $P(x_i|x_j)$, $P(x_i|\neg x_j)$, $P(x_i|x_j \wedge x_k)$, $P(x_i|x_j \wedge \neg x_k)$, $P(x_i|\neg x_j \wedge x_k)$ and $P(x_i|\neg x_j \wedge \neg x_k)$, respectively. The probability bounds $l$ and $u$ were selected uniformly at random between 0 and 1 such that $u - l \leq 0.6$, and we ensured that all instances with $n \leq 10$ were consistent.

Table 1 summarizes the results obtained for `maximax` MAP queries on the random LCNs.

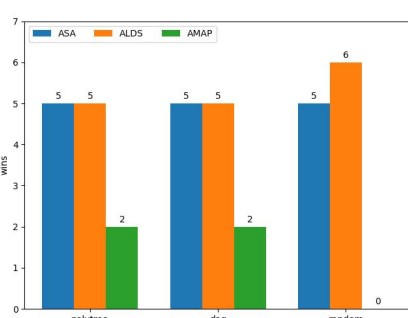

Figure 2: Wins for LCNs with $n = 10$.

For each problem class we consider both smaller ($5 \leq n \leq 10$) and larger ($30 \leq n \leq 70$) scale instances, respectively. We report the average CPU time in seconds and number of problem instance solved (out of 10) for each problem size. A '-' indicates that the respective algorithm exceeded the 2 hour time limit. The maximum discrepancy value use by algorithms LDS and ALDS was set to $\delta = 3$, while algorithms SA and ASA used up to 30 flips over a single iteration. We can see that the algorithms using exact MAP assignment evaluations (i.e., DFS, LDS and SA) are limited to small scale problem instances with up to 8 propositions and they run out of time on the larger instances. This is caused by the prohibitively large computational overhead associated with the exact evaluation of the MAP assignments during search. In contrast, the approximate search algorithms ALDS and specially ASA can scale to much larger problem instances due to the less expensive approximate MAP assignment evaluations. AMAP is the best performing algorithm in terms of running time and number of problems solved for all reported problem sizes. However, since the solution found by AMAP is only a local maxima, in Figure 2 we report on the solution quality found by algorithms AMAP, ALDS and ASA on LCN instances of size 10. Specifically, we show the number of wins as the number of times (out of 10) each algorithm found the best solution. In this case, algorithms ALDS and ASA were initialized with the MAP assignment

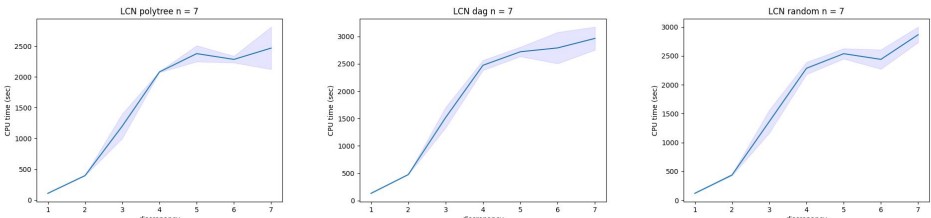

Figure 3: Average CPU time in seconds and standard deviation vs discrepancy $\delta$ for ALDS($\delta$).

Table 2: Results for MMAP tasks on realistic LCNs. CPU time in seconds. Time limit is 2 hours.

| LCN | exact MAP eval | | | | approx MAP eval | |
|---|---|---|---|---|---|---|
| | DFS | LDS(3) | SA | AMAP | ALDS(3) | ASA |
| Toy | 2.20 | 3.18 | 1.85 | 0.85 | 134.83 | 141.17 |
| Earth | 9.19 | 7.67 | 2.75 | 1.28 | 150.99 | 162.35 |
| Cancer | 16.34 | 14.09 | 8.52 | 2.64 | 157.92 | 159.66 |
| Asia | 811.82 | 800.18 | 312.10 | 4.07 | 187.44 | 201.76 |
| Credit | - | 6719.30 | 2976.55 | 5.09 | 204.77 | 222.52 |
| Engine | 4786.12 | 4502.34 | 2033.77 | 6.57 | 212.61 | 235.70 |
| Suicide | - | - | - | 5.99 | 220.31 | 203.68 |
| Tank | - | - | - | 8.04 | 263.65 | 281.73 |
| Alarm | - | - | - | 4.28 | 216.19 | 186.67 |
| Hepatitis | - | - | - | 8.22 | 260.38 | 250.45 |

found by AMAP. We can see that almost always the search-based approaches ALDS and ASA are able to find better solutions than AMAP. This is important in practice, particularly on larger scale problems where we can use AMAP to find a MAP solution quickly, and subsequently refine that solution using a search-based algorithm like ALDS or ASA if the time budget allows it. Finally, in Figure 3 we show the impact of the maximum discrepancy value $\delta$ on the running time of algorithm ALDS($\delta$). It is easy to see that as the discrepancy value $\delta$ increases, the search space explored by ALDS($\delta$) becomes larger, and therefore its corresponding running time increases as well.

**Realistic LCNs.** We experimented with a set of more realistic LCNs which were first introduced in [18]. These LCNs were derived from real-world Bayesian networks [23] and contain up to 10 propositions as well as up to 24 sentences of the form $l \leq P(x_i) \leq u$ and $l \leq P(x_i|\pi_i) \leq u$, respectively, where $x_i$ is the positive literal of proposition $X_i$ and $\pi_i = y_{i1} \wedge \cdots \wedge y_{ik}$ is the conjunction of the positive or negative literals corresponding to a particular configuration of the parents $\{Y_{i1}, \ldots Y_{ik}\}$ of $X_i$ in the Bayesian network. The specification of these LCNs is included in the supplementary material. Table 2 reports the results obtained on 10 LCN instances for the maximax MMAP task with 4 MAP propositions selected randomly. As before, algorithms DFS, LDS(3) and SA which rely on exact evaluations of the MAP assignments during search can only solve the smallest problem instances within the 2 hour time limit. In contrast, algorithms ALDS(3) and ASA solve all problem instances due to a much reduced overhead associated with the approximate MAP assignment evaluations. In this case, the search spaces explored by ALDS(3) and ASA are approximately the same in size and therefore the corresponding running times are comparable. AMAP is the fastest algorithm in this case as well.

**Application to Factuality in Large Language Models.** We consider an application of MMAP inference in LCNs to assess the factuality of the output $A$ generated by a large language model (LLM) in response to a user query $Q$ with respect to an external source of knowledge $C$ that may contain contradicting factual information (e.g., Wikipedia) [24]. The goal is to compute a *factuality score* for response $A$, denoted by $f_C(A)$, in the context of the information from $C$. In the following, we assume that $A$ can be decomposed into a set of $n$ *atomic facts* (or just *atoms*) $A = \{A_1, \ldots, A_n\}$ (e.g., one way to do that is to split $A$ into sentences) and, for each atom $A_i$, up to $k$ relevant passages $\{C_{i1}, \ldots, C_{ik}\}$ called *contexts* can be retrieved from $C$. A natural language inference (NLI) classifier such as SBERT [25] can be used to infer the *entailment*, *contradiction* and *neutrality* relationships between the texts corresponding to the atoms and contexts together

Table 3: Results for factuality LCNs. Average CPU time in seconds and number of problem instances solved. Time limit is 2 hours.

| size | exact MAP eval | | | | approx MAP eval | |
|---|---|---|---|---|---|---|
| $n, k = 2$ | DFS | LDS(2) | SA | AMAP | ALDS(2) | ASA |
| 2 | 56.95 (10) | 57.37 (10) | 60.09 (10) | 0.31 (10) | 5.25 (10) | 4.13 (10) |
| 4 | - | - | - | 0.98 (10) | 80.07 (10) | 54.15(10) |
| 6 | - | - | - | 1.97 (10) | 453.88 (10) | 219.57 (10) |
| 10 | - | - | - | 7.33 (10) | 2713.90 (10) | 928.28 (10) |
| 20 | - | - | - | 28.42 (10) | - | 3809.23 (10) |
| 50 | - | - | - | 379.18 (10) | - | - |
| 100 | - | - | - | 1807.10 (10) | - | - |

with their corresponding probabilities (or scores). Specifically, we consider relationships between an atom and a context $r(A_i, C_{ij})$, and between two contexts $r(C_{ij}, C_{pq})$, respectively, where $r \in \{\text{entailment}, \text{contradiction}\}$. We define an LCN $\mathcal{L}$ containing $n + n \times k$ propositions for each of the atoms and contexts, and two types of sentences corresponding to the entailment and contradiction relationships as follows: $l \leq P(Y|X) \leq u$ if $X$ entails $Y$, and $l \leq P(\neg Y|X) \leq u$ if $X$ contradicts $Y$, where $X$ and $Y$ are the propositions corresponding to a context and an atom, or to two different contexts, respectively. The lower and upper bounds $l$ and $u$ can be calculated easily from the probabilities obtained by running multiple NLI classifiers. Finally, the factuality score $f_C(A)$ is the proportion of true atoms in the MAP assignment obtained by solving a `maximax` MMAP task over $\mathcal{L}$ where the MAP propositions are those corresponding to $A$'s atoms.

Table 3 displays the results obtained on randomly generated factuality LCNs. More specifically, for each reported problem size $n \in \{2, 4, 6, 10, 20, 50, 100\}$, we generated 10 random instances with $n$ atoms and $k = 2$ contexts per atom such that $10\%$ of all possible pairwise relationships between atoms and contexts were selected to be either *entailment* or *contradiction* with probability $0.5$ while the remaining relationships were labeled as *neutral* and thus ignored. The lower and upper probability bounds $l$ and $u$ in the corresponding LCN sentences were also generated randomly between 0 and 1 such that $u - l \leq 0.6$. In this case, the maximum discrepancy value was set to 2 and simulated annealing was allowed a single iteration and 30 flips. We observe again that algorithms DFS, LDS(2) and SA can only solve the smallest instances due to large computational overhead associated with exact evaluation of the MAP assignments. In contrast, algorithms ALDS(2) and ASA which rely on less expensive approximate evaluations of the MAP assignments can scale to larger problems with up to 20 atoms. Algorithm AMAP outperforms its competitors and solves all problem instances.

In summary, our empirical evaluation showed that the exact search-based MAP/MMAP algorithms are limited to solving relatively small problem instances. In contrast, the approximate MAP/MMAP schemes based on either message-passing or search can scale to much larger LCN instances.

## 5 Conclusions

In this paper, we address abductive reasoning tasks such as generating MAP and Marginal MAP (MMAP) explanations in Logical Credal Networks (LCNs), a recently introduced probabilistic logic framework for reasoning with imprecise knowledge. Since an LCN encodes a set of distributions over its interpretations, a complete or partial explanation of the evidence (i.e., a MAP assignment) may correspond to more than one distribution. Therefore, we define the maximin/maximax MAP and MMAP tasks for LCNs as finding complete or partial MAP assignments that have maximum lower/upper probability given the evidence. We propose several search algorithms that combine depth-first search, limited-discrepancy search or simulated annealing with exact evaluations of the MAP assignments using marginal inference for LCNs. We also develop an approximate message-passing scheme as well as extend limited discrepancy search and simulated annealing to use an approximate evaluation of the MAP assignments during search. Our experiments with random LCNs and LCNs derived from realistic use-cases demonstrate conclusively that the search methods based on exact evaluations of the MAP assignments are limited to small size problems, while the approximation schemes can scale to much larger problems. For future work we plan to investigate more advanced depth-first branch-and-bound and best-first search techniques. However, these kinds of methods require developing novel heuristic bounding schemes to guide the search more effectively [16].

**Acknowledgements**

Fabio Cozman thanks CNPq (grant 305753/2022-3) and the Center for AI at Universidade de São Paulo, funded by FAPESP (grant 2019/07665-4) and IBM.

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
