# OpenReview forum: "Abductive Reasoning in Logical Credal Networks"
_NeurIPS.cc/2024/Conference — NeurIPS 2024 poster_

### Official Review · Reviewer_ZTSx · 2024-07-04

**Soundness:** 4
**Presentation:** 3
**Contribution:** 3
**Rating:** 7
**Confidence:** 3

**Summary:**

This paper addresses abductive reasoning tasks such as generating MAP and Marginal MAP (MMAP) explanations in Logical Credal Networks (LCNs). Given that LCNs encode sets of distributions over their interpretations, a complete or partial explanation of the evidence may correspond to multiple distributions. Thus, the authors define the maximin/maximax MAP and MMAP tasks for LCNs as finding complete or partial MAP assignments with maximum lower/upper probability given the evidence. They propose several search algorithms that combine depth-first search, limited-discrepancy search, or simulated annealing with exact evaluations of MAP assignments using marginal inference for LCNs. Additionally, they develop an approximate message-passing scheme and extend limited discrepancy search and simulated annealing to use approximate evaluations of MAP assignments during the search. Experiments show that the approximation schemes which they have proposed can scale to much larger problems compared to search methods.

**Strengths:**

1. The research is very detailed, providing an excellent formalization of abductive reasoning in LCNs. This formalization addresses a significant gap in previous LCN research, where solving MAP was challenging, and developing corresponding algorithms is not a trivial extension. The authors list several algorithms for solving this and successfully implement and compare them in detail.
2. The approximate algorithms proposed by the authors significantly improve both the solving time and the scale of solvable problems compared to previous search-based methods.

**Weaknesses:**

The experimental setup is relatively simple, lacking more practical industrial examples and relying more on basic toy experiments. Additionally, there is no comparison with other methods.

**Questions:**

1. In the initial LCN papers, the authors compared LCN with ProbLog and MLN. Can your method be compared with these? For example, ProbLog is a direct derivative of logic programming, which can perform basic reasoning, including deductive reasoning as well as abductive reasoning.
2. In the experiments, different approximate algorithms show significant differences in solving time, despite having almost equivalent time complexity. How can this be explained?
3. In line 80, should it be "less likely"?

**Limitations:**

Yes.

---

> ### Author Rebuttal · Authors · 2024-08-02
>
> We thank the reviewer for their valuable feedback. We provide responses to your questions and concerns below.
>
> We would like to emphasize that our paper provides the first study dedicated to MAP and MMAP inference in LCNs and to the best of our knowledge there are no other baseline algorithms for solving these tasks in LCNs.
>
> Q1: Previous work on LCNs showed that the Problog/MLN formalisms cannot really be used to model the same benchmark problems that LCNs can model (for instance, Problog/MLN do not allow conditional probability bounds). Therefore, a direct comparison with these methods is not really possible on the benchmark problems we consider in our work.
>
> Q2: The discrepancies between LDS/SA and ALDS/ASA in terms of running times can be explained by the slightly different search spaces explored by the two classes of algorithms. LDS/ALDS explore up to 2^d nodes, where d is the discrepancy value, and every single assignment is evaluated from scratch. In contrast SA/ASA are limited to M=30 flips (i.e., assignments) in our experiments, but some of these assignments may be generated multiple times and, in this case, SA and ASA use caching, namely if the current MAP assignment was solved before, they retrieve its value from the cache, thus avoiding its re-evaluation. Therefore, SA/ASA are more efficient than LDS/ALDS and we demonstrate this in our experiments. Furthermore, the underlying ipopt solver we use to solve the non-linear programs corresponding to the conditioned subproblems often suffers from numerical precision issues and for some subproblems it may take longer to solve than for others, and consequently the order in which these conditioned subproblems are considers may impact the overall running time. We will discuss these aspects in more detail in the paper.
>
> Q3: Yes, it was supposed to be “less likely” – we will correct the typo, thanks for the catch.

---

> > ### Comment · Reviewer_ZTSx · 2024-08-10
> >
> > Thank the authors for their response. I appreciate the clarification, especially regarding the runtime, which has addressed my concerns. However, I still believe that comparing your method with approaches outside of LCN, even if those methods use the most naive or brute-force algorithms, is necessary. Nonetheless, I consider this paper to be solid work and will maintain my current rating.

---

### Official Review · Reviewer_56vh · 2024-07-09

**Soundness:** 4
**Presentation:** 2
**Contribution:** 2
**Rating:** 5
**Confidence:** 2

**Summary:**

This paper proposes how to solve MAP and Marginal MAP queries for Logical Credal Networks (LCNs). LCNs are a class of graphical probabilistic logic models with the expressiveness to represent cycles as well as marginal and conditional probability bounds on logical formulae. The authors first present three search algorithms for exactly computing MAP/MMAP bounds on queries to LCNs. Then, because the MAP/MMAP problem for LCNs is NP-Hard, the authors present three approximation methods that offer considerable speedup at only a small cost to accuracy. The supplemental material contains formal proofs and extensive experiment details.

**Strengths:**

The authors do a good job of comprehensively studying the problem of MAP/MMAP inference for LCNs. Consequently, this paper is relevant and useful for researchers and practitioners interested in graphical probabilistic logic models. In particular:
* The authors present a variety of different algorithms for MAP/MMAP on LCNs and theoretically prove their correctness and complexity.
* The approximation algorithms presented offer considerable speedup while achieving competitive performance with the exact solutions.
* The six algorithms (three exact + three approximate) offer good coverage of what a user of LCNs may consider.

**Weaknesses:**

My primary critique is on the exposition and motivation.
* It would help to better motivate the strengths of LCNs by providing examples/references of why cycles and marginal+conditional probabilities show up. This would be stronger than claiming "... which may be important in many realistic use cases".
* Having more examples and figures of LCNs would be helpful, especially in Section 2.1.
* Emphasizing the NP-Hardness of MAP/MMAP for LCNs would help better motivate the need for search techniques and approximation algorithms.
* Section 2.2 is a bit dense. Moreover, it appears that Equation (8) is a vector-valued objective, which does not make sense to me. Overall, this section would benefit from a more relaxed exposition pace --- possibly in a future manuscript version.

The experiments would also benefit from having the main takeaways more explicitly stated.
* It would help to have captions that succinctly explain the main points, e.g., for Table 1: that AMAP does very well compared to DFS, LDS(3), and SA.
* It would be useful to supplement Figure 2 with a plot of the optimality gaps of the approximations, rather than simply the "wins".

Others comments:
* At present, it appears that the practical algorithms are restricted to fairly small LCNs.
* Three exact algorithms and three approximation schemes in one paper are quite a lot. It would help the reader to see the author's recommendations and discussions of their various trade-offs.
* The part on "Application to Facuality in Large Language Models" is a bit dense and sudden. If the authors deem it within the scope of the paper, it would help to supplement this with some experiments.

Minor Comments:
* Section 3.2: It would be helpful to explicitly list Algorithm 1, 2, and 3 in the paragraph headers, e.g., "Algorithm 1: Depth-first Search"
* More descriptive names for theorem labels would be helpful, e.g. "Theorem 1 (Complexity of Depth-first Search)"

**Questions:**

* The space complexities appear quite extreme. Could the authors please comment on whether this is a fundamental drawback, and if so, how much better might one (in practice) reasonably expect to do?
* Figure 3 in the paper looks different from Figure 6 in the supplemental material. Could the authors please address why this might be the case?


I have saved my harshest critiques for last. I am an outsider to the graphical models community, so please forgive my ignorance. I am willing to revise my assessment if the authors could please expand on the following points that a general ML audience might have:
* Why care about LCNs?
* What are real-world cases of people using LCNs or similar models?
* What's an example of something that LCNs can capture that other graphical models can't?
* In fact, why care about graphical models at all when deep learning is everywhere? (Some comments about explainability + interpretability should be well-received)
* If the existing algorithms for LCNs are so expensive, what would be some "realistic" scenarios for which they're relevant?

**Limitations:**

The authors sufficiently discuss the limitations of their work.

---

> ### Author Rebuttal · Authors · 2024-08-02
>
> We thank the reviewer for their valuable feedback. We provide responses to your questions and concerns below.
>
> We will expand the background section to include more examples of LCNs. For now, we refer the reader to the previous work on LCNs which we cite in the paper. We agree that Section 2.2 is quite dense at the moment and we will use additional space to add more details as well as a small running example. The hardness of inference in LCNs is not actually known yet. We suspect it belongs to the NP^NP^PP-hard class but proving the result formally is an open problem. We will include a short discussion to emphasize this issue. Finally, we will follow up on your suggestion and summarize the findings of our empirical evaluation in a separate subsection.
>
> Q1: Regarding space complexity, yes, in the worst case we need to represent in memory a probability distribution over an exponentially large number of interpretations. This is a serious limitation, especially for exact algorithms. For example, in practice, our approximate scheme AMAP can handle LCN instances whose factor graph contains factor nodes involving up to 10-12 propositions.
>
> Q2: Figure 3 in the main paper plots results with algorithm ALDS and is the same as Figure 5 in supplementary material (although there is a typo in its caption), while Figure 6 in supplementary material contains results with algorithm LDS.
>
> Q4: Previous work on LCNs has already showcased several potential applications of LCNs including one from the chemistry domain. In this paper, we illustrate a potential application to factuality assessment for LLMs. Furthermore, [Cozman et al, 2024] has shown recently that LCNs can be used to model and solve causal reasoning tasks such as estimating the causal effect of an intervention under partial identifiability conditions.
>
> Q3&Q5: LCNs allow specifying conditional probability bounds on logic formulae and allow directed cycles. Previous work on LCNs demonstrated that this is very useful especially when we need to combine multiple sources of imprecise knowledge in a single model. In contrast, graphical models like Bayes nets do not allow cycles nor bounds on probability values, credal networks allow probability bounds but require acyclicity. Probabilistic logics like Problog and MLNs allow undirected cycles but require point probability values. Therefore, LCNs can be viewed as a generalization of these previous models.
>
> Q6: Graphical models are inherently interpretable models and can be used to provide explanations in a principled manner. They have been studied extensively over the past decades and there is substantial literature illustrating non-trivial applications to real-world situations.
>
> Q7: We show that exact inference for LCNs is expensive while approximation schemes are far more scalable. These approximate algorithms are clearly applicable to the potential realistic applications presented in previous papers on LCNs (please see also our answer to Q4).
>
> We thank the reviewer again for their detailed feedback and thoughts. If the reviewer believes we have addressed some of their concerns, we request them to consider increasing their score.

---

> > ### Comment · Reviewer_56vh · 2024-08-11
> > **Reply to authors**
> >
> > Thank you for the detailed response. I am still skeptical of LCNs due to their currently limited adoption, but I am warming up to their potential usefulness. I have increased my score.

---

### Official Review · Reviewer_Pbqt · 2024-07-10

**Soundness:** 3
**Presentation:** 3
**Contribution:** 3
**Rating:** 6
**Confidence:** 4

**Summary:**

This is a paper about inference on logical credal networks, a class of graphical models that cope with interval-valued probabilistic statements on propositional logic formulae. The novelty of the paper is that it focuses on marginal MAP inference (and hence also MAP as a special case). Exact and approximate algorithms based on search strategies are proposed and empirically tested.

**Strengths:**

The main contribution is an approximate procedure based on recent work on marginal inference in the same class of models. This procedure seems to perform well on models for which exact algorithms are too slow. The extension from marginal inference to marginal MAP is non-trivial.

The experiments are extensive and the results convincing.

**Weaknesses:**

LCNs are not very popular models, at least for the moment, and their potential for applications to real problems is not very clear.
Something similar could be said, more specifically, to the need for tools for abductive reasoning in such models.

**Questions:**

- The authors do not report results on the hardness of their inference tasks. This is probably obvious, right? Yet, some comments about that would help. It would be also interesting to compare a brute-force wrt the exact methods.
- With credal models, there is a difference between the "conditional" and the "joint" versions of an inference task for the simple reason that the two models are proportional through the probability of the evidence, which is not constant in a credal setup. I believe that the authors consider a "joint" version, but this point should be made clearer.
- I don't understand how the ground-truth values in the experiments are obtained.
- The results show that the topology of the networks seems not to affect the execution time too much. Some comments about that would be valuable, as the situation is very different in other graphical models.

**Limitations:**

-

---

> ### Author Rebuttal · Authors · 2024-08-02
>
> We thank the reviewer for their valuable feedback. We provide responses to your questions below.
>
> Q1: The hardness of MAP/MMAP inference in LCN isn’t actually known yet. We suspect it is an NP^NP^PP-hard task but proving this result formally is an open problem. We will include a discussion of this issue. The DFS algorithm described in the paper (Algorithm 1) is a brute-force algorithm (i.e., the MAP assignments are enumerated exhaustively and each one is evaluated exactly) so we do include results with such a brute-force approach.
>
> Q2: Yes, the MAP/MMAP tasks defined in this paper can be viewed as “joint” inference tasks. They find a truth assignment to the MAP propositions and the evidence propositions can be viewed as part of that assignment. We will clarify this in the paper.
>
> Q3: We are not clear what the reviewer means by “ground-truth values”. If they refer to the exact MAP/MMAP solutions, we do obtain them with the DFS algorithm which is an exact algorithm but only on the smallest problem instances due to the scalability issues discussed in the paper. However, if they refer to the probability values in the problem instances considered, we actually generated those values randomly as described in the experimental section (and also in the supplementary material).
>
> Q4: That is correct. The algorithms proposed in this paper do not exploit the graph structure as commonly done in variational inference in graphical models. Understanding the factorization of the LCN is an open research problem. Recently, [Cozman et al, 2024] investigated Markov conditions and factorization in LCNs which could be used in principle to develop more efficient algorithms for LCNs. This ambitious endeavour is also part of our research agenda.

---

> > ### Comment · Reviewer_Pbqt · 2024-08-12
> >
> > I thank the authors for their comments. All my questions/doubts have been clarified, and I am happy to confirm my positive opinion about that paper.

---

### Official Review · Reviewer_AMr2 · 2024-07-12

**Soundness:** 3
**Presentation:** 2
**Contribution:** 2
**Rating:** 5
**Confidence:** 3

**Summary:**

The paper presents an approach for MAP and marginal MAP in logical credal networks using search based algorithms. Compared to PGMs, MAP and marginal MAP is harder in LCNs since a MAP assignment could correspond to one of several distributions and therefore, to even evaluate the MAP, we need to perform marginalization which is a hard task. The exact algorithms are developed using DFS, limited discrepancy search and simulated annealing. However, since these are infeasible in practice, approximate methods are developed based on a marginal inference method for LCNs. The main idea is to compute lower and upper MAP (or marginal MAP) probabilities

Experiments are performed on synthetic LCNs and those generated from Bayesian nets. Further, an application related to testing LLMs is presented. Specifically, the idea is to connect atoms from the generated text to a source such as wikipedia and compute facility as a the MAP score. Scalability results are shown for the LCNs used in this task for different variants of the proposed methods.

**Strengths:**

- A novel class of inference queries added to LCNs can improve the applicability of LCNs in different applications.
- The paper develops a comprehensive suite of MAP and MMAP exact and approximate inference algorithms for LCNs
- Results show that the proposed algorithms can scale up and find approximate MAP/MMAP solutions

**Weaknesses:**

In terms of significance of the proposed approaches, the results mainly show scalability of the approximate methods and their ability to find MAP/MMAP solutions. However, the actual application of MAP/MMAP seems missing. For example, in the LLM application the results do not really tell us how useful the MAP/MMAP solution was compared to other methods other than that the MAP/MMAP solution could be found as the LCN becomes larger. In general, I feel the proposed approaches would be much more significant if the results showed that if the proposed solution improved over other approaches that could be used to solve the same problem.

**Questions:**

Are there specific use cases where the MAP or MMAP solutions for LCNs can be compared with other competing methods? In general, what would be the advantages of MAP/MMAP queries for LCNs as compared to other probabilistic approaches.

**Limitations:**

Limitations are not explicitly mentioned.

---

> ### Author Rebuttal · Authors · 2024-08-02
>
> We thank the reviewer for their valuable feedback. We provide responses to your questions below.
>
> To the best of our knowledge, this paper provides the first study dedicated to MAP and MMAP inference in LCNs and therefore, there are no other baseline algorithms for solving MAP/MMAP queries in LCNs to compare with. In general, LCNs offer several advantages over existing models, namely they allow specifying probability bounds on logic formulae, do not require acyclicity, and are far more flexible to specify logic formulae compared with existing logic programming approaches.
>
> MAP/MMAP queries for LCNs provide a principled way to generate most probable (partial) explanations for these kinds of models. For example, inferring the code in the uncertain Mastermind puzzles introduced in a previous paper on LCNs can be solved as a MMAP query and as shown previously the most effective way to solve it is by modelling the problem as an LCN rather than using existing approaches based on Bayes nets, Problog or MLN.

---

> > ### Comment · Reviewer_AMr2 · 2024-08-12
> >
> > Thanks for your response. I think the issue of LCNs being able to solve problems better than other statistical relational models is good. I do think that being the first work to add MAP and MMAP (both of which are important tasks in graphical models) to LCNs is a valuable contribution. The technical components seem solid in the work in terms of improving scalability of MAP/MMAP queries in LCNs, but I was still a bit unsure about the quality of the approximation algorithms relative to some other approach, particularly since I think it is hard to have theoretical guarantees on the approximation. That would have made the paper much stronger in terms of the significance I feel. In summary though, I feel this seems like a solid enough work with some weaknesses.

---

### Official Review · Reviewer_NnQo · 2024-07-13

**Soundness:** 3
**Presentation:** 2
**Contribution:** 3
**Rating:** 6
**Confidence:** 3

**Summary:**

Logical Credal Networks (LCNs) are a probabilistic logic framework designed for representing and reasoning with imprecise knowledge. While previous research on LCNs has focused on marginal inference, there has been a lack of exploration in abductive reasoning within this context. This paper addresses this gap by investigating abductive reasoning in LCNs using Maximum A Posteriori (MAP) and Marginal MAP (MMAP) queries. To solve MAP and MMAP tasks, the authors propose techniques based on Depth-First Search (DFS), Limited Discrepancy Search (LDS), and Simulated Annealing (SA). Additionally, to improve time complexity, they introduce a method that utilizes a message-passing approximation scheme, allowing for more efficient and scalable solutions.

**Strengths:**

- The paper introduces a novel method for computing MAP and MMAP in Logical Credal Networks (LCNs), which were not previously addressed in this context. It proposes a more efficient search method compared to the brute force approach for MAP calculation.
- Unlike previous studies, this research reduces complexity by using an approximation method to address time complexity.
- This approach is theoretically well-proven and demonstrated through experimental results.
- By adding a simple method called Message Parsing Approximation to LDS and SA, performance was effectively enhanced.
- The practical application of the algorithm is shown, proving its usefulness and suggesting future development directions.
- Since this paper shows that LCN's knowledge expression is better than other existing research, many follow-up studies using LCN could be conducted.

**Weaknesses:**

Comparison with prior work & Experiments
- Mention how you build upon from the previous works by Radu Marinescu et al. The introduction mentions that existing approaches use heuristic algorithms or DP algorithms for MAP and MMAP inference, and this paper does not employ a significantly different method.
- If this is a follow-up to the aforementioned research, wouldn't it have been better to demonstrate the performance of the previous research with the addition of Message Passing Approximation? The current paper compares the method added to LDS and SA. The advantages of using SA and LDS compared to previous research are not clearly demonstrated.
- While I agree that there may not be existing studies that have introduced LCNs, there are certainly prior studies that have tackled MAP and Marginal MAP tasks. It would be beneficial to include baselines comparing the performance of the proposed algorithms with those of existing studies. Without such baselines, the current experimental results only allow for comparisons among the proposed algorithms themselves, making it difficult to ascertain whether these algorithms are superior to those from other research. The absence of baseline comparisons hampers a complete understanding of the contribution of the proposed methods.
- A comparison and introduction of MAP estimation methods in Credal Networks (CN) and Bayesian Networks (BN) would have been beneficial to understand the practical advantages over these existing methods. (Probably author thought this was the scope of the prior work - Logical Credal Network)
- It is unclear whether ALDS and ASA are needed instead of AMAP. Experimental results show that AMAP consistently outperforms in terms of CPU time(in Tables 1, 2, 3), and the auxiliary measure of solved problem instances(in Tables 1, 3) is always 10/10 with AMAP. These results raise questions about the necessity of ALDS and ASA. Although the paper attempts to address these concerns with Figure 2, but it demonstrates that sufficient problem-solving can be achieved without the optimal LCN, suggesting that the approximation methods of ALDS and ASA may be inefficient.

Presentation
- There is insufficient evidence for the statement in LINE 24: "Logical Credal Networks (LCNs) have focused exclusively on marginal inference, i.e., efficiently computing posterior lower and upper probability bounds on a query formula."
- The main contribution seems to be Algorithm 4, but its emphasis is not different from Algorithms 1-3, which perform inference through DFS, making it difficult to identify the core of the paper.
- Line 36: Is the term "probability bound" more accurate than "imprecise probability."?
- Section 3 lack sufficient explanation of the proposed methodology, making it difficult for readers to fully understand. Additionally, the paper lacks a clear analysis of the time complexity and space complexity of the algorithms, which is crucial for evaluating their efficiency.

**Questions:**

- Could you provide more detailed descriptions of the algorithm implementation? Additionally, could you include a clear analysis of the time and space complexity of the algorithms to help evaluate their efficiency? I recommend this paper should answer these questions in the Appendix. (Could be in the supplementary material)
- Could you include baselines comparing the performance of the proposed algorithms with those of existing studies? Without such baselines, it is challenging to determine whether your algorithms are superior to existing research.
- Could you clarify the specific advantages of ALDS and ASA over AMAP, and provide additional justification for their inclusion? How do these algorithms contribute to this paper?

**Limitations:**

- As the authors mentioned, the method still relies on heuristic approaches for finding MAP, indicating a need for further development. The proposed method is not yet practical for real-world applications without additional improvements, as evidenced by the computational overhead and limited scalability shown in the experiments.
- Arbitrary Setting of Hyperparameter Values:
   * Size of problem (n): Figure 3 shows a trend where CPU time increases as the discrepancy increases for n = 7. However, Table 1 shows experiments conducted for n = 5, 8, 10, 30, 50, 70 for the same datasets. It is unclear why only Figure 3 uses other values, indicating an arbitrary setting of hyperparameters without a specific criterion.
   * Discrepancy (delta): Table 1 and Table 2 use a discrepancy value of 3, whereas Table 3 uses a discrepancy value of 2 for the experiments. The rationale behind these hyperparameter settings needs to be clarified. Setting discrepancy values without a consistent criterion undermines the study's consistency and can confuse interpreting the results. Furthermore, a discrepancy of 2 appears to be an elbow point, suggesting it is an optimal parameter value for ALDS(2). Maybe, results for the large data(n = 30, 50, 70) could have been obtained using ALDS(2).
   * Contexts per atom (k): In Table 3, since k is fixed to 2, you don't have to indicate repeatedly it in the table. Also, it would be better to append the results of experiments with various k.
- Uncertain Contribution: While this paper proposes various algorithms based on LCN, it is questionable whether the introduction of LCN is necessary for solving Marginal MAP. I summarize the points previously mentioned from this perspective. Firstly, in the performance of the approximation algorithms, it is evident that AMAP, which relatively fails to find the optimal LCN, performs better. This result weakens the necessity of LCN to solve the marginal MAP problem. Additionally, it is difficult to prove the superiority of LCN without a theoretical comparison (e.g., time complexity, space complexity) or experimental comparison (i.e., experimental results) between LCN-based algorithms and existing algorithms. To prove the contributions of this paper, it seems necessary to supplement these aspects.

---

> ### Author Rebuttal · Authors · 2024-08-02
>
> We thank the reviewer for their valuable feedback. We provide responses to your questions and concerns below.
>
> The MAP and MMAP inference tasks have been extensively studied over the past decades in the context of classical graphical models such as Bayesian networks or Markov networks. However, algorithms developed for those models such as variable elimination or depth-first AND/OR branch and bound are not directly applicable to LCNs and therefore a direct comparison with those kinds of methods is not possible. For example, the AND/OR search algorithms for MMAP in Bayes nets presented in [Marinescu et al, JAIR-2018] are guided by a heuristic function derived from a variational bound on MMAP in Bayes nets. That bound cannot be computed in an LCN and therefore the AND/OR search algorithms mentioned simply do not work on LCNs.
>
> Previous papers on LCN have only focused on exact and approximate algorithms for computing upper and lower probability bounds on a query formula (also known as marginal inference). Virtually nothing is known about MAP and MMAP inference in LCNs. Therefore, the contribution of our paper is to bridge this gap and present the first ever exact and approximate algorithms for solving MAP and MMAP queries in LCNs.
>
> Regarding the experiments, in Tables 1, 2 and 3 we chose a discrepancy value such that the search space explored by LDS/ALDS would have a comparable size to that of SA/ASA. However, we experimented with many more discrepancy values (up to 7) but observed that a larger discrepancy value has a negative impact on running time and we illustrate this behavior with Figure 3.
>
> Q1: We included the actual Python implementation of the proposed algorithms in the supplementary material (see the exact_map.py and approx_map.py scripts) and we are currently in the process of open sourcing our code. Theorems 1, 2, 3 and 4 in the main paper provide the time and space complexity bounds of the proposed algorithms. Their proofs are included in the supplementary material.
>
> Q2: To the best of our knowledge, our paper provides the first study on MAP and MMAP inference in LCNs and therefore there are no other baseline algorithms to compare with on these two tasks.
>
> Q3: Algorithms ALDS and ASA could potentially improve the solution found by AMAP. More specifically, the initial solution found by AMAP is most likely a local optima, but if more time is available then we can use ALDS/ASA to search for a better solution. Figure 2 in the main paper is meant to illustrate the benefit of using ALDS/ASA on top of AMAP. In this case, both ALDS and ASA were initialized with the solution found by AMAP and the plot shows how many times ALDS/ASA found a better solution compared with the initial one upon exceeding the time limit. We will expand the discussion in the paper to emphasize the benefits of ALDS/ASA over AMAP.

---

> ### Comment · Reviewer_NnQo · 2024-08-11
>
> Thank you for answering my questions.
>
> As another reviewer commented, it would be beneficial to initiate a discussion with the audience on how we can incorporate the idea of symbolic reasoning (LCN) in the era of large neural networks. Potentially, neural networks could provide some unknown LCN sentences that were missed by domain experts, making the overall framework more comprehensive.
>
> As the last reviewer (ZTSx) mentioned, comparing your method with approaches outside of LCN could be helpful to justify that LCN is also practically useful among diverse solutions. Additionally, providing more discussion on the benefits of ALDS/ASA over AMAP would help in the detailed understanding of the methods.
>
> Although I am not an expert in the field, I believe this paper is beneficial in advancing research towards building a System 1+2 engine. I look forward to seeing more contributions in neuro-symbolic abductive reasoning. I will maintain my positive score.
>
> I would like to suggest lowering the entry barrier of this manuscript for audiences who aren't familiar with LCN and MMAP. For instance, including toy examples of MAP and MMAP in medical diagnosis or fault detection could effectively illustrate how these inference tasks are applied in real-world scenarios. While the supplementary material includes LCNs for multiple real-world scenarios, some readers may also require background knowledge on MAP and MMAP. Consider reorganizing the manuscript and supplementary materials to enhance readability.
>
> -----
>
> These are still a few remaining questions, so I would like to ask for some additional clarification.
>
> > Q1. Could you provide more detailed descriptions of the algorithm implementation? Additionally, could you include a clear analysis of the time and space complexity of the algorithms to help evaluate their efficiency? I recommend this paper should answer these questions in the Appendix. (Could be in the supplementary material)
>
> > A1. We included the actual Python implementation of the proposed algorithms in the supplementary material (see the exact_map.py and approx_map.py scripts) and we are currently in the process of open sourcing our code. Theorems 1, 2, 3 and 4 in the main paper provide the time and space complexity bounds of the proposed algorithms. Their proofs are included in the supplementary material.
>
> We feel that an additional explanation in the proof would be beneficial for clarity. For instance, in the proof of Theorems 1, 2, and 3, it is mentioned that the complexity is O(2^2^n) because the LCN has 2^n interpretations, but this point could be elaborated further to ensure a clear understanding. Regarding the proof of Theorem 4, the claim that "the complexity of algorithm AMAP is dominated by the complexity of the factor-to-node messages" could benefit from additional clarification. Specifically, it would be helpful to show the time complexities of the other parts of algorithms to support this claim.
>
> > Q2. Could you include baselines comparing the performance of the proposed algorithms with those of existing studies? Without such baselines, it is challenging to determine whether your algorithms are superior to existing research.
>
> > A2. To the best of our knowledge, our paper provides the first study on MAP and MMAP inference in LCNs and therefore there are no other baseline algorithms to compare with on these two tasks.
>
> We are curious about the practical benefits of using the proposed MAP/MMAP algorithms with LCNs compared to Bayesian networks and Credal Networks. In scenarios where LCNs are expected to be advantageous due to the richer information they encode, it would be helpful to know the extent of these benefits. Additionally, since LCNs use more information, is there a concern regarding potential overfitting?
>
>
> > Q3. Could you clarify the specific advantages of ALDS and ASA over AMAP, and provide additional justification for their inclusion? How do these algorithms contribute to this paper?
>
> > A3. Algorithms ALDS and ASA could potentially improve the solution found by AMAP. More specifically, the initial solution found by AMAP is most likely a local optimum, but if more time is available then we can use ALDS/ASA to search for a better solution.
> Figure 2 in the main paper is meant to illustrate the benefit of using ALDS/ASA on top of AMAP. In this case, both ALDS and ASA were initialized with the solution found by AMAP and the plot shows how many times ALDS/ASA found a better solution compared with the initial one upon exceeding the time limit. We will expand the discussion in the paper to emphasize the benefits of ALDS/ASA over AMAP.
>
> You mentioned that AMAP is likely to get stuck in local optima, and we would like to understand the reason behind this. Additionally, we are interested in learning more about the specific situations where ALDS and ASA perform better relative to each other. Insight into the conditions under which one algorithm outperforms the other would be valuable.

---

### Author Rebuttal · Authors · 2024-08-02

We would like to thank all reviewers for their valuable feedback and thoughtful suggestions.

Since several reviewers have asked for more justification for the formalism, we would like to emphasize that LCNs offer a language to deal with many AI settings where probabilities and constraints interact and that they are meant to go beyond classical graphical models such as Bayesian networks and the like. For instance, LCNs offer a path to dealing with non-identifiability in causal reasoning as was recently shown by [Cozman et al, 2024]. Furthermore, LCNs can offer a path to uncertainty quantification where it is important to differentiate between epistemic and aleatoric uncertainties, something that is done often with probability bounds [Hullermeier et al, 2022]. Unlike existing graphical models (e.g, Bayes nets) and probabilistic logics (e.g., Problog, MLN), LCNs are more expressive allowing conditional probability bounds on logic formulae, do not require acyclicity restrictions, and in general are more flexible to specify logic formulae compared with existing formalisms. Previous work on LCNs has already showcased several applications that can be solved more efficiently when modeled as LCNs and where existing approaches based on graphical models like Bayes nets or on probabilistic logics like Problog and MLN fail. These applications include uncertain Mastermind puzzles, credit card fraud detection with imprecise domain expert knowledge, as well as an application from the chemistry domain involving a prediction task using imprecise domain expert knowledge and molecular fingerprinting data [Marinescu et al, 2022, 2023]. Clearly, identifying additional real-world applications for LCNs is an open problem and is also part of our ongoing research agenda.

References:

[Cozman et al, 2024] F. Cozman, R. Marinescu, J. Lee, A. Gray, R. Riegel, D. Bhattacharjya. Markov Conditions and Factorications in Logical Credal Networks. In International Journal of Approximate Reasoning (IJAR), vol. 172, 2024.

[Hullermeier et al, 2022] E. Hullermeier, S. Destercke, and M. Shaker. Quantification of Credal Uncertainty in Machine Learning: A Critical Analysis and Empirical Comparison. Proceedings of the 38th Conference on Uncertainty in Artificial Intelligence (UAI), 2022.

---

### Decision · Program_Chairs · 2024-09-25

**Decision:**

Accept (poster)

**Comment:**

In the family of probabilistic graphical models, logical credal networks (LCNs) have recently garnered attention for their ability to represent imprecise probabilities in a compact and intuitive way. Up to now, there has been limited exploration of queries for this model class. This paper aims to broaden the practical applications of LCNs by focusing on the widely used maximum a posteriori (MAP) and marginal MAP queries. After providing an overview of LCNs and their queries, the authors propose exact and approximate algorithms for MAP and marginal MAP inference. These algorithms are evaluated using both synthetic and real-world benchmarks.

All reviews for this paper are positive, ranging from borderline accept to accept. As acknowledged by reviewers, this paper provides the first approach to handle abductive reasoning (with MAP and marginal MAP) on logical credal networks. As the runtime (and space) complexity of exact inference algorithms is prohibitive, an important contribution of this paper is to focus on the empirical analysis of various approximation algorithms, including message-passing, limited discrepancy search, and simulated annealing. In summary, this paper paves the way for future research on abductive reasoning with imprecise graphical models.

While the main concerns of reviewers have been addressed in the fruitful discussions with authors, I highly recommend providing a light revision of the current manuscript to consider all reviewers' comments. Notably:
* Applicability: as several reviewers were skeptical about the practical interest of LCNs, the insightful global response provided by the authors should be incorporated into the paper.
* Complexity: the paper should include a discussion on the computational complexity of MAP (and marginal MAP) for LCNs in order to justify the use of approximation algorithms. I would suggest providing an NP^PP hardness result (even if, as suspected by the authors, the complexity class is even higher). By first taking a class of undirected or directed graphical models for which MAP is NP^PP hard and then explaining that this class can be transformed (in polynomial time and space) into an LCN, the hardness result should follow.
* Presentation: The paper, in its current state, is targeted to specialists in probabilistic graphical models. The too short introduction should be revised to motivate this study for a larger NeurIPS audience. Furthermore, in order to deal with the density of the paper, the authors’ suggestion to provide additional examples and use additional space would be welcome. Finally, even if the state-of-the-art methods for (M)MAP inference are dedicated to probabilistic models and cannot be trivially extended to credal models, this should be emphasized at the beginning of the experimental section.